# String Stable Control of Connected Vehicles via Multi-Agent Lyapunov Actor-Critic

## Abstract

Networked or interconnected systems, such as urban transportation networks, rely on robust control strategies to ensure string stability-the concept that prevents the amplification of disturbances through the network. This capability is critical for system performance. A directly motivated example is the mitigation of phantom traffic jams, where string stability can suppress stop-and-go wave propagation. In this paper, we first establish a sufficient condition for scalable input-to-state stability (sISS), providing a theoretical guarantee for string stability. The derived conditions reveal a coupled shrinking relationship among the energy of different agents in the system, which depend on local Lyapunov functions but guarantee the global condition of sISS. Based on this theoretical foundation, we propose a practical and effective algorithm, named multi-agent Lyapunov actor-critic (MALAC), to achieve stable control in networked systems. Numerical simulations demonstrate that MALAC can ensure the string stability in the cooperative adaptive cruise control task.

## 1 Introduction

Interconnected/networked systems have wide applications in different fields, including transportation systems Chu et al. (2020), vehicle controls Li et al. (2020), electronic power grid Mahela et al. (2022), and robotic swarms Antonelli (2013). The string stability of interconnected systems, as defined by Swaroop & Hedrick (1996), ensures uniform boundedness of the state of the systems. This property prevents the amplification of disturbances and a consequent deterioration of performance across the network, which is particularly crucial for large-scale systems.

A prime and highly representative example for the string stability of interconnected systems is the car-following problem. In such a setting, each vehicle adjusts its velocity based on the behavior of the preceding vehicle, forming a dynamical chain of interconnected agents. While individual stability ensures safe tracking of an immediate leader, it does not prevent disturbances (e.g., sudden braking and acceleration) from amplifying along the vehicle platoon. This string-unstable phenomenon may cause oscillations to grow downstream, potentially leading to traffic jams, increased energy consumption, or even accidents. Consequently, ensuring string stability is crucial in designing control strategies for car-following systems, particularly for connected and automated vehicles.

Due to its rapid development and superior performance, reinforcement learning (RL) Sutton & Barto (2018) has been widely used to solve the control problem of connected systems, including traffic signal control Chen et al. (2020) and cruise control Desjardins & Chaib-draa (2011); Chu et al. (2020). The majority of existing work predominantly focuses on local stability Berkenkamp et al. (2017); Chow et al. (2018), with only a limited subset addressing string stability, and even those lack theoretical guarantees. On the one hand, the stability of the individual subsystem is not sufficient for the safe operation of the network Silva et al. (2025). On the other hand, works on reward shaping Peake et al. (2020), action filtering Liu et al. (2024), and state enhancement Jiang et al. (2025) cannot provide theoretical insight to achieve the string stability.

In this paper, we try to investigate the notion of **scalable input-to-state stability (sISS)** in control theory Besselink & Knorn (2018) and design an RL algorithm to achieve the string stability of the interconnected systems. One recent work Zhou et al. (2025) designs an RL-based controller that can provide string stability guarantees. However, the designed verification and synthesis framework relies on external verification tools, which are widely adopted methodologies. In comparison, we try

to design an algorithm originating from the stability theory, which eliminates the need for external tools.

The contributions of this paper are threefold. First, we establish a sufficient condition for scalable input-to-state stability, providing a theoretical guarantee for string stability. The derived conditions reveal a coupled shrinking relationship among the energy of different agents in the system, which depend on local Lyapunov functions but guarantee the global condition of sISS. Second, based on this theoretical foundation, we propose a practical and effective algorithm, named multi-agent Lyapunov actor-critic (MALAC), to achieve stable control in networked systems. It utilizes the Lyapunov critic and string stability theory to facilitate the policy optimization. Third, we design the RL-based car-following environments based on the well-known transportation simulator, SUMO Lopez et al. (2018), to evaluate the performance of our proposed algorithm.

## 2 RELATED WORK

**Stability and reinforcement learning.** There are many different definitions of stability in control theory, from classical Lyapunov stability Khalil & Grizzle (2002) to string stability Swaroop & Hedrick (1996); Feng et al. (2019). In this paper, we aim to ensure the string stability of the interconnected systems, which highlights the importance of suppressing disturbance amplification.

While reinforcement learning (RL) has shown great potential in many control tasks Chen et al. (2020); Zhang et al. (2025), some try to introduce the stability notion into RL algorithm design for safety and robustness purposes. Many efforts have introduced Lyapunov-based methods to embed stability guarantees into RL Berkenkamp et al. (2017); Chow et al. (2018); Han et al. (2020), yet these approaches are primarily limited to single-agent systems. In other words, these works seek to be *locally* stable instead of string stable. Local stability guarantees that small perturbations around an equilibrium for an individual agent decay Zhou et al. (2025), intuitively being more like a concept of "convergence". Instead, string stability prevents the amplification of perturbations through the network. It is more like a concept of "network performance or robustness" Besselink & Knorn (2018). Thus, local stability alone is insufficient for string stability.

**String-stable system control.** One of the most representative string-stable control tasks is cruise control (or vehicle control). Well-known vehicle control models include Intelligent Driving Model (IDM) Treiber et al. (2000), Adaptive Cruise Control (ACC) Ploeg et al. (2011), and Cooperative Adaptive Cruise Control (CACC) Milanés & Shladover (2014). While ACC is already available in commercial markets, it suffers from the obvious string instability. CACC utilizes the Vehicle-to-Vehicle (V2V) communications to reduce the instability to some extent Milanés & Shladover (2014); Lei et al. (2012). As a result, many works try to further facilitate the CACC with RL techniques Desjardins & Chaib-draa (2011); Chu et al. (2020). However, most prior RL-based approaches improve empirical performance through reward shaping or heuristic state/action designs Desjardins & Chaib-draa (2011); Peake et al. (2020); Liu et al. (2024); Jiang et al. (2025), which do not ensure string stability. Our approach contributes to this line by proposing a multi-agent Lyapunov actor–critic algorithm (MALAC), which unifies control-theoretic guarantees with multi-agent RL and achieves stable coordination in transportation networks. More recent work on verification frameworks with string stability guarantee leverages neural certificates but requires external tools detached from policy learning Zhou et al. (2025). In contrast, our work directly integrates stability conditions into the learning process via Lyapunov critics, providing theoretical guarantees without additional verification.

## 3 PROBLEM STATEMENT AND PRELIMINARY

### 3.1 PROBLEM STATEMENT

We formalize our problem into a multi-agent system. Consider an interconnected system of $N$ agents, denoted as the set $\mathcal{N} = \{1, \cdots, N\}$. The connection topology of agents is represented by an adjacency matrix $G \in \{0, 1\}^{N \times N}$, where each element $G_{i,j} = 1$ if agent $j$ is coupled to agent $i$ and $G_{i,j} = 0$ otherwise. Let $\mathcal{N}_i = \{j | G_{i,j} = 1\} \subseteq \mathcal{N}$ denote the set of neighbors of agent $i$. $x_i \in \mathcal{X}_i$ represent the state of agent $i$, and $d_i$ is the external disturbance affecting agent $i$. As

formalized in the field of control, the dynamics model of agent $i \in \mathcal{N}$ is a differential equation

$$\dot{x}_i = f_i(x_i, \{x_j\}_{j \in \mathcal{N}_i}, a_i, d_i) \tag{1}$$

where $\dot{x}_i$ is the time derivative and $a_i$ is the control input (or action) of agent $i$.

As such networks grow, they might suffer from a form of instability that can cause the amplification of disturbances through the network and deterioration of performance, regardless of whether the agents themselves are stable Silva et al. (2025). As a result, in this paper, we try to develop an effective multi-agent reinforcement learning algorithm to mitigate the string instability of the interconnected systems.

*Motivated Example.* Imagine a platoon of vehicles. The state of each vehicle can be described by its distance relative to the preceding vehicle and the ego speed. The control input is the acceleration, which is used to adjust the vehicle's speed. Thus, the vehicle dynamics are defined by the rate of change of distance and speed. Now, suppose the lead vehicle suddenly encounters a bumpy section of road and chooses to decelerate rapidly to reduce the bumps. The following vehicles, in order to avoid a collision, also begin to decelerate sequentially. If the vehicles in the following queue accelerate too abruptly, their speed may drop below that of the lead vehicle (so-called speed undershoot Lei et al. (2012)). To catch up with the lead vehicle, the slower vehicles may need to accelerate again to match the lead vehicle's speed. This can generate stop-and-go waves that propagate in the opposite direction of the platoon. This phenomenon is known as "phantom traffic" Knorr et al. (2012), which is frequently observed in everyday life.

## 3.2 STRING STABILITY DEFINITION

Many string stability definitions have been studied, please refer to the survey Feng et al. (2019) for differences among different definitions. In this paper, we investigate scalable input-to-state stability Besselink & Knorn (2018), as stated in Def. 1.

**Definition 1.** *(Scalable Input-to-state Stability) The system Eq. 1 is sISS if there exists a class-$\mathcal{KL}$ function $\beta$ and a class-$\mathcal{K}_\infty$ function $\gamma$ such that, for any $N \in \mathbb{N}$, any initial conditions $x_i(0)$, and any disturbance $d_i$, the inequality*

$$\max_{i \in \mathcal{N}} |x_i(t)|_2 \leq \beta(\max_{i \in \mathcal{N}} |x_i(0)|_2, t) + \gamma(\max_{i \in \mathcal{N}} \|d_i\|_{\mathcal{L}_\infty}) \tag{2}$$

*holds for $\forall t \geq 0$ and $i \in \mathcal{N}$.*

where $|x|_2$ is vector 2-norm and $\|x\|_{\mathcal{L}_\infty}$ is the shorthand notation of the signal norm $\|x(t)\|_{\mathcal{L}_\infty}^{\mathcal{T}} = \sup_{t \in \mathcal{T}} |x(t)|_\infty$. A continuous function $\gamma : [0, a) \to [0, \infty)$ is a *class-$\mathcal{K}$ function* if it is strictly increasing and $\gamma(0) = 0$. It is said to belong to *class-$\mathcal{K}_\infty$* if $a = \infty$ and $\gamma(r) \to \infty$ as $r \to \infty$. A continuous function $\gamma : [0, a) \to [0, \infty)$ is a *class-$\mathcal{L}$ function* if it monotonically decreases to 0 as its argument tends to $\infty$. A function $\beta : [0, a) \times [0, \infty) \to [0, \infty)$ is a *class-$\mathcal{KL}$ function* if, for fixed $t \geq 0$, $\beta(\cdot, t)$ is a class-$\mathcal{K}$ function and, for fixed $r \geq 0$, $\beta(r, \cdot)$ is a class-$\mathcal{L}$ function.

This stability definition has several benefits. First, compared to other string stability, which is typically restricted to unidirectional structures (e.g., vehicle platoons), sISS generalizes the concept to arbitrary network topologies Besselink & Knorn (2018). It is actually an extension to the string stability. Second, in Def. 1, the functions $\beta$ and $\gamma$ are the same for any $N$. In other words, the upper bound of the state norm $\max_i |x_i(t)|_2$ is independent of the number of systems $N$, so it is scalable. Last but not least, it puts focus on stopping disturbances from spreading, giving a measure of robust performance instead of just stability.

Since we have a useful definition of the string stability, the next step is to design an effective multi-agent reinforcement learning algorithm to achieve this stability.

## 4 DISCRETE sISS SUFFICIENT CONDITION

Since most of reinforcement learning algorithms are designed on a discrete time system and for practical use consideration, we first discretize the dynamics model 1 over a sampling time interval $\Delta t$. Note that, discretization is also widely used in the field of control for developing practical

algorithms Zhang et al. (2025). Then, by applying a difference approximation of the derivative, we can rewrite the dynamics model as

$$x_i(k+1) = x_i(k) + \Delta t \cdot f(x_i(k), \{x_j(k)\}_{j \in \mathcal{N}_i}, a_i(k), d_i(k)) \tag{3}$$

where $k \in \mathbb{N}^+$. The terms $x_i(k)$, $a_i(k)$, and $d_i(k)$ represent the state, action, and the disturbance of agent $i$ in the $k$-th time-step, respectively.

Based on the discrete-time system in Eq. 3, we prove a sufficient condition (i.e., Theorem 1) to achieve the sISS, which is also the theoretical foundation of our multi-agent algorithm design.

**Theorem 1.** *Consider a network of systems of the form 3. Assume that for agents $i \in \mathcal{N}$, there exists a local positive definite function $V_i : \mathbb{R}^{n_i} \to \mathbb{R}_+$, which verifies $\alpha_1(|x_i|_2) < V_i(x_i) < \alpha_2(|x_i|_2)$ for some class-$\mathcal{K}_\infty$ functions $\alpha_1$ and $\alpha_2$, and such that, for all $(x_j)_{j \in \mathcal{N}_i} \in \prod_{j \in \mathcal{N}_i} \mathbb{R}^{n_j}$, the inequality*

$$V_i(k+1) \le \sum_{j \in \mathcal{N}_i} c_{i,j} V_j(k) + \Delta t h_i |d_i(k)|^2 \tag{4}$$

*holds for any trajectory that is consistent with the dynamics 3, with $1 \ge c_{i,j} \ge 0$, and $h_i \in \mathbb{R}$ is a constant that weights the action of the external disturbance applied to the $i$-th agent. $V_i(k)$ is the abbreviation of $V_i(x_i(k+1))$. $\mathcal{N}_i$ only contains the connected neighbors of agent $i$, not including itself. If the constants $c_{i,j}$ are such that the following condition is satisfied, for some $c > 0$:*

$$c\Delta t + \sum_{j \in \mathcal{N}_i} c_{i,j} \le 1, \ \forall i \in \mathcal{N} \tag{5}$$

*then the equilibrium point of the system is sISS in the sense of Def. 1 and $\mathcal{V} = [V_1, \cdots, V_N]^\top$ is an sISS vector Lyapunov function.*

*Proof.* The detailed proof can be found in Appendix. A.2. $\square$

If we consider the function $V_i$ as a representation of agent $i$'s energy (as classical Lyapunov function has always been seen as the energy function of a system), this theorem indicates a coupled shrinking relationship among the energy of different agents. It also guides a possible way to design the MARL algorithm. Eq. 4 indicates that, the next-step energy of any agent $i$ is no larger than the weighted sum of the energy of its neighbors and the disturbance. Once the situation in Eq. 5 is satisfied, it tells that $\sum_{j \in \mathcal{N}_i} c_{i,j}$ is strictly smaller than 1, which pushes the energy of agent $i$ into a lower level. As a result, since inequality holds for any agent in the system, the energy of the whole multi-agent system can converge and thus be stable.

Specifically, if we rewrite the above inequality into a matrix form, we can obtain

$$\mathcal{V}(k) \le C\mathcal{V}(k-1) + \Delta t H D(k-1) \le C^2 \mathcal{V}(k-2) + \Delta t C H D(k-2) + \Delta t H D(k-1)$$
$$\cdots$$
$$\le C^{k+1} \mathcal{V}(0) + \Delta t \sum_{m=0}^{k-1} C^m H D(k-1-m) \tag{6}$$

Here, the diagonal elements of matrix $C \in \mathbb{R}^{N \times N}$ are all zeros, while the off-diagonal elements satisfy $1 > c_{i,j} > 0$, and are zero otherwise. Matrix $H \in \mathbb{R}^{N \times N}$ is a diagonal matrix with $h_i$ as its diagonal entries, and $D(k)$ denotes the disturbance vector at step $k$. It can be observed in Eq. 6 that, since every element of $C$ is less than 1, as $k$ increases, $C^k$ inevitably converges to the zero matrix. This implies that the vector Lyapunov function $\mathcal{V}(k)$ will also converge to the zero vector. Moreover, the influence of external disturbances occurring in the past will diminish over time. Therefore, the entire system tends toward stability.

In addition, Eq. 5 also reveals the relationship between the time interval $\Delta t$ and the coefficient $c_{i,j}$. The smaller the discrete time step, the easier it is for Eq. 5 to be satisfied, and consequently, Theorem 1 is more readily established. Conversely, a larger time interval may make it difficult to find a control policy that ensures system stability.

It is worth noting that, this sufficient condition is directly inspired by Theorem 1 in Silva et al. (2025). Their theory is designed for the continuous-time system, and a recent work directly adopts

it to realize string-stability control Zhou et al. (2025). In comparison, our Theorem 1 is in a simpler and intuitive form, and also provides the connection to the sampling time interval $\Delta t$ in a discrete-time system. These insights and characteristics can facilitate the practical and effective MARL algorithm design.

# 5    PRACTICAL ALGORITHM DESIGN

In this section, we propose a multi-agent actor-critic algorithm to learn stability-guaranteed policies. First, we introduce the Lyapunov critic function. Then, based on the maximum entropy actor-critic framework, we use the Lyapunov critic function and the sISS sufficient condition in presenting our MARL algorithm.

## 5.1    LYAPUNOV CRITIC FUNCTION

In control tasks, there is always a cost function $c(x, a)$ to measure how good or bad a state-action pair is. In the RL framework, this cost function plays a central role in guiding the agent's behavior. Specifically, the agent interacts with the environment by selecting actions based on its current state, and the cost provides immediate feedback on the quality of those decisions. To evaluate not only the instantaneous performance but also the long-term consequences of actions, RL introduces the concept of a value function. Among them, the critic function is often represented as the state-value function $V(x)$ or the action-value function $V(x, a)$. Here, $V(x) = \mathbb{E}_{a \sim \pi(\cdot|x)}[V(x, a)]$ where $\pi$ is the control policy. The critic function serves as an approximation of the expected cumulative cost.

There is an interesting fact that value functions in RL are Lyapunov functions if the costs are strictly positive away from the origin Berkenkamp et al. (2017). Following the existing work Han et al. (2020), we construct the Lyapunov critic for each agent $i \in \mathcal{N}$ with the following parameterization:

$$V_i(x, a) = f_{\phi_i}(x, a)^\top f_{\phi_i}(x, a) \tag{7}$$

where $f_{\phi_i}$ of agent $i$ is the output vector of a fully connected neural network with parameter $\phi_i$. This parameterization ensures the positive definiteness of $V_i(s)$, which satisfies the basic property of $V_i$ in Theorem 1.

Moreover, $L_i$ is updated to minimize the following objective function:

$$J_i(V_i) = E_{\mathcal{D}} \left[ \frac{1}{2} (V_i(s, a) - V_i^{\text{target}}(s, a))^2 \right] \tag{8}$$

Here, $V_i^{\text{target}}$ is the Lyapunov candidate and acts as an approximation target. We define $V_i^{\text{target}}$ as the value function:

$$V_i^{\text{target}}(s, a) = c(s, a) + \gamma \mathbb{E}_{s' \sim p(\cdot|s, a)}[V_i'(s')] \tag{9}$$

where $p(\cdot|s, a)$ is the state transition probability and $V_i'$ is the target network parameterized by $\phi'$. This update method is widely adopted in RL algorithms Haarnoja et al. (2018). $V_i'$ has the same structure as $V_i$, but the parameter $\phi'$ is updated through exponentially moving average of the weights of $V_i$ controlled by a hyperparameter $\tau$, $\phi'_{k+1} \leftarrow \tau \phi_k + (1 - \tau) \phi'_k$.

## 5.2    MULTI-AGENT LYAPUNOV ACTOR-CRITIC

Based on the Lyapunov critic function and Theorem 1, we present a novel MARL algorithm, called Multi-Agent Lyapunov Actor-Critic (MALAC), to achieve the string-stable control. For each agent $i \in \mathcal{N}$ in the interconnected system, the policy learning problem is summarized as a constrained optimization problem:

$$\text{find } \pi_{\theta_i} \tag{10}$$

$$\text{s.t. } E_{\mathcal{D}} \left[ V_i(s') - \sum_{j \in \mathcal{N}_i} c_{i,j} V_j(s) + \epsilon \right] \leq 0 \tag{11}$$

$$\sum_{j \in \mathcal{N}_i} c_{i,j} < 1, \quad c_{i,j} \geq 0 \tag{12}$$

$$- E_{\mathcal{D}} \left[ \log(\pi_{\theta_i}(a|s)) \right] \geq \mathcal{H}_t \tag{13}$$

where the control policy $\pi_{\theta_i}$ is parameterized by a deep neural network $f_{\theta_i}$. The constraint 11 is the core condition in Theorem 1. The difference with Eq. 4 is that we use a hyperparameter $\epsilon > 0$ to replace $-\Delta t h_i |d_i|^2$. It is because the disturbance $d_i$ is always unpredictable. Eq. 13 is the minimum entropy constraint borrowed from the maximum entropy RL framework to improve the exploration Haarnoja et al. (2018), and $\mathcal{H}_t$ is the target entropy.

One remaining question is how to decide the value of $c_{i,j}$. Here, we introduce a simple method called equal potential assignment, i.e., let $c_{i,j} = \frac{1}{|\mathcal{N}_i|+1}$ which satisfies $0 < \sum_{j \in \mathcal{N}_i} c_{i,j} < 1$. Then, by introducing the Lagrangian method, the optimization object is

$$J(\theta_i) = \mathbb{E}_{(s,a,s') \sim \mathcal{D}} \left[ \alpha \log(\pi_{\theta_i}(a|s)) + \lambda (V_i(s') - \frac{1}{|\mathcal{N}_i|+1} \sum_{j \in \mathcal{N}_i} V_j(s) + \epsilon) \right] \quad (14)$$

where $\alpha$ and $\lambda$ are Lagrange multipliers. Under this assignment, agent $i$ takes equal consideration of its connected neighbors, which is especially intuitive and reasonable in a homogeneous networked system. And also, this method is easy to implement and can still achieve the desired string stability, as shown in the experiments. Besides, the parameters of the policy network are update through stochastic gradient descent of Eq. 14.

## 6 EXPERIMENTS

### 6.1 EXPERIMENTAL SETUP

We conduct experiments on a representative task, cooperative adaptive cruise control (CACC), to verify the effectiveness of the proposed method. We build a new simulation environment based on the well-known transportation simulator Simulation of Urban Mobility (SUMO) Lopez et al. (2018). SUMO provides diverse control functions and facilitates flexible modifications, including creating different road networks, setting different vehicle types, etc. SUMO also provides multiple classical car-following models, including Intelligent Driving Model, the Adaptive Cruise Control model, the Cooperative Adaptive Cruise Control model, etc. It is worth noting that, ACC is already adopted by commercial cars but may cause the string instability Milanés & Shladover (2014).

In our experiments, we simulate a string of 4 vehicles for 100s on a single lane road, with a 0.1s control interval. The speed of the leading car (so-called leader) is controlled manually to simulate the scenarios of "Catch-up" and "Slow-down", which is a common setting to test the cruise control task Chu et al. (2020); Zhou et al. (2025). Specifically, we design a new scenario called "Wave". The leader keeps the speed of 20m/s in the first 10 seconds, then it accelerates to 25m/s with the acceleration of 2m/s$^2$. This is the "Catch-up" phase. It keeps driving with 25m/s for 43s and deaccelerate back to 20m/s with the acceleration of -2m/s$^2$. This is the "Slow-down" phase. For the followers, they start with the initial speed of 20m/s powered by the basic ACC model. Once all followers appear on the road, we turn off the control of ACC and let our MALAC agents control the acceleration of the followers. The action, observation, and cost (i.e., reward) are set as follows:

- **Action**: The action of each agent $i$ is the acceleration $a_{t,i} \in [-2.5, 2.5]$. Existing RL-based cruise control studies only provide 3 or 4 discrete action candidates instead of a continuous action space Chu et al. (2020); Desjardins & Chaib-draa (2011), which may not be practical in real life. In comparison, our continuous control task could be more challenging.

- **Observation**: The observation $x_{t,i} \in S_i$ includes the headway (spacing) $h_{t,i}$, the ego speed $v_{t,i}$, and the speed difference, i.e., $x_{t,i} = (h_{t,i}, v_{t,i}, v_{t,i-1} - v_{t,i})$. The information on headway and speed difference is always available by using V2V communication and radar sensors Milanés & Shladover (2014). Besides, to implement our algorithm, we also assume that the agent can access the critic value of its predecessor for policy optimization.

- **Cost**: The cost function is $c_{t,i} = (h_{t,i} - h^*)^2 + (v_{t,i} - v_{t,i-1})^2 + a_{t,i}^2$. $h^*$=20m is the target headway. This form of reward is simple and intuitive. The first term regarding headway is to keep a safe distance. The second term regarding velocity is for efficiency consideration. The last term takes both safety and comfort into consideration to avoid large acceleration. Besides, when a collision happens, a large cost of 500 is assigned. Similar reward design can be found in existing works Chu et al. (2020); Chen et al. (2024); Liu et al. (2024).

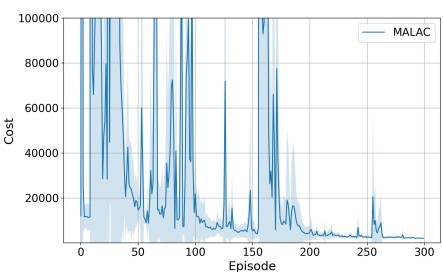

Figure 1: Total episode cost of three follow-
ers. The shaded region represents the stan-
dard deviation of the average evaluation over
5 runs.

Table 1: Training results on 300th episode:
Average speed and average headway with
their standard deviation over 5 runs.

| Foll. | Avg. Speed | Avg. Headway |
|---|---|---|
| 1 | 22.35 ($\pm$ 0.001) | 20.06 ($\pm$ 0.137) |
| 2 | 22.31 ($\pm$ 0.002) | 20.01 ($\pm$ 0.174) |
| 3 | 22.26 ($\pm$ 0.003) | 20.03 ($\pm$ 0.108) |

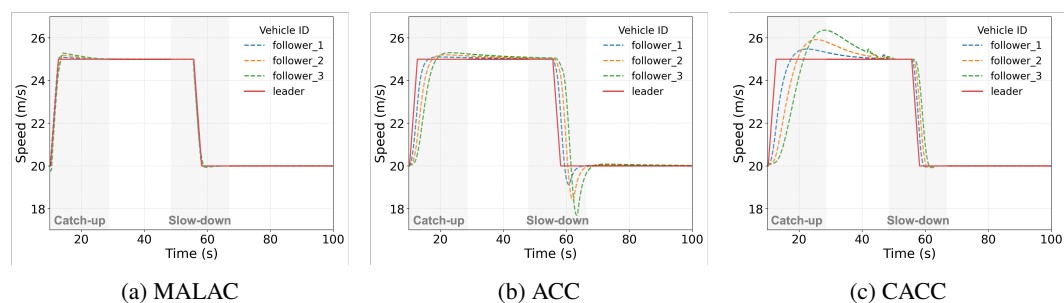

(a) MALAC          (b) ACC          (c) CACC

Figure 2: Vehicle's speed over time under different car-following models.

The initial data is collected by using the default ACC model in SUMO. The model is trained in a
central training and decentralized execution (CTDE) style. Agents share the same actor and critic
networks. The number of episodes of 1 training round is 300, and there are 1000 steps in one
episode. The training results are averaged over 5 runs. Experiments were conducted on a MacBook
Pro equipped with an Apple M1 Pro chip (8-core CPU, 14-core GPU, 16GB RAM).

## 6.2 TRAINING RESULTS

Fig. 1 illustrates the trend in the cumulative cost of three followers, representing the learning curve.
As shown in the figure, the learning curve exhibits a noticeable convergence after 200 episodes.
Table 1 illustrates the average speed and average headway on the 300th episode. Recall that, the
cost includes a speed-aware term $(v_{t,i} - v_{t,i-1})^2$ and a headway-aware term $(h_{t,i} - h^*)^2$. As
shown in the table, the speeds of the three followers are very close, indicating the driving efficiency.
The average headway is also close to the target headway $h^* = 20$m. These results show that the
algorithm can converge and the controlled vehicles can drive as expected.

## 6.3 PERFORMANCE COMPARISON

Fig. 2 illustrates the vehicle speeds during the driving process for MALAC, ACC, and CACC. As
mentioned before, string instability refers to the phenomenon in which external disturbances con-
tinuously propagate and amplify within a vehicle platoon. In the context of car-following scenarios,
this implies that a sudden acceleration or deceleration of the lead vehicle results in increasingly pro-
nounced speed fluctuations among the following vehicles. As illustrated in Fig. 2, both the ACC and
CACC car-following models implemented in SUMO exhibit evident manifestations of such instabil-
ity. Specifically, the ACC model demonstrates substantial velocity oscillations under the slow-down
condition, whereas the CACC model shows pronounced fluctuations in the catch-up scenario. In
contrast, our proposed method substantially mitigates these instability effects.

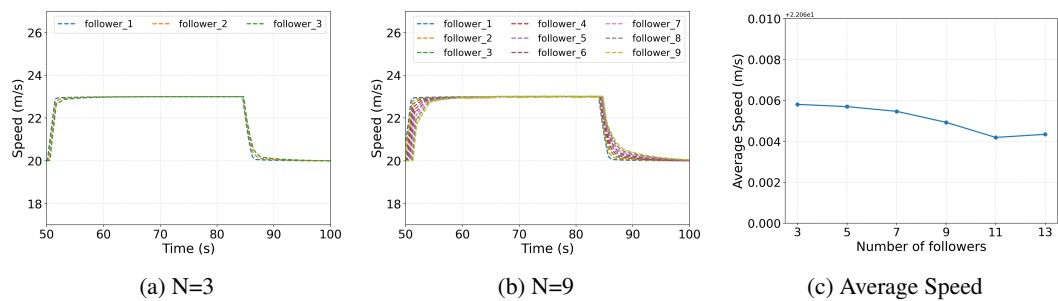

(a) N=3        (b) N=9        (c) Average Speed

Figure 3: The generalization results on **Highway**: the leader's speed changes from 20m/s to 23m/s.

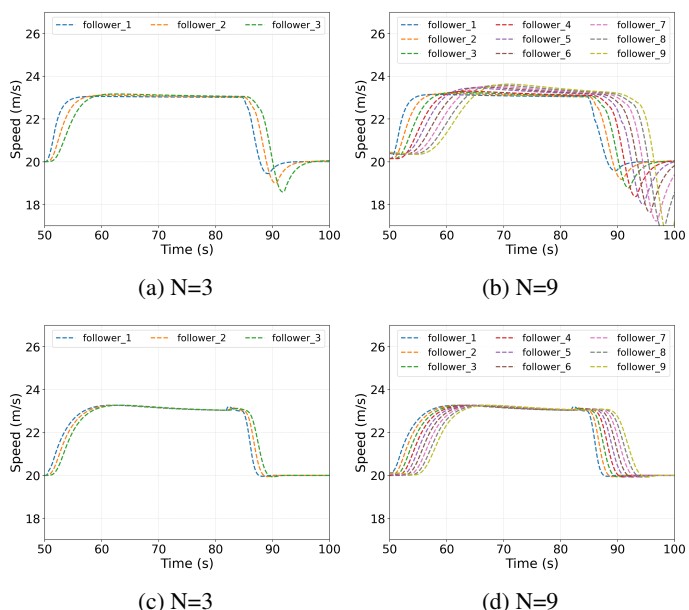

(a) N=3                 (b) N=9

(c) N=3                 (d) N=9

Figure 4: Results of ACC and CACC model on **Highway**

## 6.4 GENERALIZATION EXPERIMENT

To further assess the generalization capability of the algorithm, we transferred the trained model to a previously *unseen* scenario for testing. The scenario, referred to as "**Highway**", is designed to simulate driving conditions on a high-speed road with a relatively higher speed limit. In this case, the lead vehicle accelerates from 20m/s to 23m/s, maintains 23m/s for 33 seconds, and then decelerates back to 20m/s. All other simulation parameters remain consistent with the previous settings. Moreover, to investigate the relationship between instability and platoon length, we conducted experiments with the number of following vehicles ranging from 3 to 13.

The results of this set of experiments are quite interesting. First, as shown in Fig. 3a, 3b, and 3c for the Highway scenario, it can be observed that instability does not occur regardless of the number of following vehicles. Fig. 3c illustrates the relationship between the average vehicle speed and the platoon size, measured from the moment the lead vehicle begins to change its speed (i.e., at 50s) until 100s. The smaller the variations in this curve with respect to platoon size, the smaller the fluctuations in vehicle speeds across the platoon, which to some extent reflects system stability. Interestingly, although the variations in Fig. 3c are extremely small, the curve even exhibits a very slight downward trend. In contrast, in Fig. 4, the ACC model shows obvious instability while the CACC has a better performance than ACC.

## 7 CONCLUSION AND DISCUSSION

In this work, we investigated the problem of ensuring string stability in networked systems such as urban transportation networks. We established a sufficient condition for scalable input-to-state stability (sISS), which provides a theoretical guarantee for preventing the amplification of disturbances across agents. Building on this foundation, we developed the multi-agent Lyapunov actor-critic (MALAC) algorithm, which offers a practical and effective approach to stabilizing control in interconnected systems. Numerical simulations on the cooperative adaptive cruise control task demonstrated that MALAC successfully achieves string stability. These results suggest that the proposed framework holds promise for broader applications in large-scale networked control systems.

It is worth noting that since the concept of sISS represents a general form of stability—with desirable properties such as independence from network topology and scalability—the MALAC algorithm designed on this basis also inherits such generality, rather than being limited to the CACC task tested in this paper. In the future, MALAC could be applied to a wider range of scenarios, such as traffic signal control and drone swarm control. Moreover, it would be valuable to explore its deployment in larger-scale environments.

## REPRODUCIBILITY STATEMENT

**Reproducibility Statement.** We provide an anonymized repository containing source code, experiment scripts, and configuration files to reproduce all results: `https://anonymous.4open.science/r/MALAC1-D319`. Detailed hyperparameters, random seeds, and hardware specifications are documented to ensure reproducibility.

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

## A    APPENDIX

### A.1    USE OF LLMS

In preparing this submission, I made limited use of a large language model (GPT-5) as an assistive tool. Specifically, the model was employed for (i) generating alternative visualizations to better present experimental results, and (ii) improving the clarity and readability of the manuscript through language polishing. All research activities, including literature review, theoretical development, algorithm design, and experimental implementation and analysis, were carried out entirely by the authors. The responsibility for the accuracy and integrity of the work rests fully with the authors.

### A.2    PROOF OF THEOREM 1

Since Theorem 1 provides a sufficient condition, our objective is to prove that the theorem implies Def. 1. We first distract $\Delta t$ from the coefficient $c_{i,j}$. Let $c_{i,j} = \Delta t a_{i,j}$. Since the inequality $c\Delta t + \sum_{j \in \mathcal{N}_i} c_{i,j} \le 1$ holds for some $c > 0$, we can easily find some $c_{i,i}$ satisfying $\Delta t c + \sum_{j \in \mathcal{N}_i} c_{i,j} \le c_{i,i} \le 1$. Thus, we let $c_{i,i} = \Delta t a_{i,i}$ and obtain

$$V_i(k+1) - \sum_{j \in \mathcal{N}_i} c_{i,j} V_j(k) - \Delta t h_i |d_i(k)|^2 \tag{15}$$

$$= V_i(k+1) - V_i(k) + V_i(k) - \Delta t \sum_{j \in \mathcal{N}_i} a_{i,j} V_j(k) - \Delta t h_i |d_i(k)|^2 \tag{16}$$

$$\ge V_i(k+1) - V_i(k) + \Delta t a_{i,i} V_i(k) - \Delta t \sum_{j \in \mathcal{N}_i} a_{i,j} V_j(k) - \Delta t h_i |d_i(k)|^2 \tag{17}$$

$$\ge V_i(k+1) - V_i(k) - \Delta t \left( -a_{i,i} V_i(k) + \sum_{j \in \mathcal{N}_i} a_{i,j} V_j(k) + h_i |d_i(k)|^2 \right) \tag{18}$$

Based on the Theorem 1, we can obtain

$$V_i(k+1) - V_i(k) - \Delta t \left( -a_{i,i}V_i(k) + \sum_{j \in \mathcal{N}_i} a_{i,j}V_j(k) + h_i|d_i(k)|^2 \right) \leq 0 \tag{19}$$

Now, our target is to prove that the above inequality implies Def. 1.

By rewriting Eq. 19 in matrix form, we obtain

$$\mathcal{V}(k+1) \leq (I + \Delta tA)\mathcal{V}(k) + \Delta tHD(k) \tag{20}$$

where the diagonal element of matrix $A \in \mathbb{R}^{N \times N}$ is $-a_{i,i}$ and others are $a_{i,j}$ $(i \neq j)$. $H$ is the diagonal matrix with the diagonal element $h_i$ and the vector $D(k) = [|d_1(k)|^2, \cdots, |d_N(k)|^2]^\top$.

We then construct a comparison system $\mathcal{W}(k+1) = (I + \Delta tA)\mathcal{W}(k) + \Delta tHD(k)$ and $\mathcal{W}(0) = \mathcal{V}(0)$. We show that $\mathcal{V}(k) \leq \mathcal{W}(k)$ and then we can complete the proof by showing the boundedness of the trajectories of the comparison system.

**Lemma 1.** *Given $\Delta t \leq \frac{1}{\max_i a_{i,i}}$, let the comparison system $\mathcal{W}(k+1) = (I + \Delta tA)\mathcal{W}(k) + \Delta tHD$ and $\mathcal{W}(0) = \mathcal{V}(0)$, then $\mathcal{V}(k) \leq \mathcal{W}(k)$ holds for any $k \in \mathbb{N}^+$.*

*Proof.* We prove this lemma by induction. First, $\mathcal{V}(k) \leq \mathcal{W}(k)$ holds for the origin $k = 0$, i.e., $\mathcal{W}(0) = \mathcal{V}(0)$. Second, we assume $\mathcal{V}(k) \leq \mathcal{W}(k)$ then

$$\mathcal{V}(k+1) \leq (I + \Delta tA)\mathcal{V}(k) + \Delta tHD \leq (I + \Delta tA)\mathcal{W}(k) + \Delta tHD = \mathcal{W}(k+1) \tag{21}$$

In other words, $\mathcal{V}(k+1) \leq \mathcal{W}(k+1)$ also holds. As a result, the assumption $\mathcal{V}(k) \leq \mathcal{W}(k)$ for any $k \in \mathbb{N}^+$. $\square$

So the following part is to prove the boundedness of the trajectories of the comparison system $\mathcal{W}(k+1) = (I + \Delta tA)\mathcal{W}(k) + \Delta tHD(k)$. For simplicity, we let $\tilde{D}(k) = HD(k)$ and $B = I + \Delta tA$. By introducing the recursion, we can derive

$$\begin{aligned} \mathcal{W}(k) &= B\mathcal{W}(k-1) + \Delta t\tilde{D}(k-1) \\ &= B\left( B\mathcal{W}(k-2) + \Delta t\tilde{D}(k-2) \right) + \Delta t\tilde{D}(k-1) \\ &\cdots \\ &= B^k\mathcal{W}(0) + \sum_{j=0}^{k-1} B^j \Delta t\tilde{D}(k-1-j) \end{aligned} \tag{22}$$

Since $c_{i,i} = \Delta_t a_{i,i} \leq 1$ holds for any $i \in \mathcal{N}$, there is $\Delta t \leq \frac{1}{\max_i a_{i,i}}$. Then, assign the norm to both sides of the recursive equation and we can get

$$\|\mathcal{W}(k)\|_\infty = \left\| B^k\mathcal{W}(0) + \Delta t \sum_{j=0}^{k-1} B^j \tilde{D} \right\|_\infty \tag{23}$$

$$\leq \|B\|_\infty^k \|\mathcal{W}(0)\|_\infty + \Delta t \sum_{j=0}^{k-1} \|B\|_\infty^j \left\| \tilde{D} \right\|_\infty \tag{24}$$

$$\leq (1 - \Delta tc)^k \|\mathcal{W}(0)\|_\infty + \Delta t \sum_{j=0}^{k-1} (1 - \Delta tc)^j \left\| \tilde{D} \right\|_\infty \tag{25}$$

$$\leq e^{-k\Delta tc} \|\mathcal{W}(0)\|_\infty + \Delta t \sum_{j=0}^{k-1} e^{-j\Delta tc} \left\| \tilde{D} \right\|_\infty \tag{26}$$

$$\leq e^{-k\Delta tc} \|\mathcal{W}(0)\|_\infty + \frac{\Delta t}{1 - e^{-\Delta tc}} \left\| \tilde{D} \right\|_\infty \tag{27}$$

where $\|B\|_\infty = \max_i \sum_j |b_{i,j}|$ and $\|\mathcal{W}\|_\infty = \max_i |W_i|$. The second inequality is based on the fact that

$$\|B\|_\infty = \max_i(|1 - \Delta t a_{i,i}| + \Delta t \sum_j a_{i,j}) = \max_i(1 - \Delta t a_{i,i} + \Delta t \sum_j a_{i,j}) \leq 1 - c\Delta t$$

The third inequality comes from the fact that $1 - x \leq e^{-x}$ and the last inequality is derived based on the series convergence property.

Then, we can obatin, for each $i \in \mathcal{N}$, the inequality that upper bounds the component $W_i$ of the vector function $\mathcal{W}$ as follows:

$$\sup_i |W_i(k)|_2 \leq e^{-k\Delta t c} \sup_i |W_i(0)|_2 + \frac{\Delta t}{1 - e^{-\Delta t c}} \sup_i \left\|\tilde{d}_i\right\|_{\mathcal{L}_\infty} \tag{28}$$

Then, we use Lemma. 1, with $V_i(0) = W_i(0)$, to obtain

$$\sup_i |V_i(k)|_2 \leq e^{-k\Delta t c} \sup_i |V_i(0)|_2 + \frac{\Delta t}{1 - e^{-\Delta t c}} \sup_i \left\|\tilde{d}_i\right\|_{\mathcal{L}_\infty} \tag{29}$$

which can be written as

$$\sup_i |V_i|_2 \leq \beta\left(\sup_i |V_i(0)|_2, k\right) + \gamma\left(\left\|\tilde{d}_i\right\|_{\mathcal{L}_\infty}\right) \tag{30}$$

where $\beta(r,k) = e^{-k\Delta t c}r$ and $\gamma(r) = \frac{\Delta t}{1 - e^{-\Delta t c}}r$

Based on the increasingness of $\alpha_1$, we can write

$$\alpha_1(\sup_i |x_{t,i}|_2) = \sup_i \alpha_1(|x_{t,i}|_2) \leq \sup_i |V_i(x_{t,i})|_2 \tag{31}$$

$$\leq \beta\left(\sup_i |V_i(0)|_2, k\right) + \gamma\left(\sup_i \left\|\tilde{d}_i\right\|_{\mathcal{L}_\infty}\right) \tag{32}$$

$$\leq \beta\left(\alpha_2(\sup_i |x_i(0)|_2), k\right) + \gamma\left(\sup_i \left\|\tilde{d}_i\right\|_{\mathcal{L}_\infty}\right) \tag{33}$$

where the last inequality is using Lemma 4.3 in Khalil & Grizzle (2002).

Further, by applying the inverse of $\alpha_1$ to both sides and using the triangle inequality of the form $\alpha(a+b) \leq \alpha(2a) + \alpha(2b)$ (for any class-$\mathcal{K}$ function $\alpha(\cdot)$, see Eq. (12) in Sontag et al. (1989)), we can obtain:

$$\sup_i |x_{t,i}|_2 \leq \alpha_1^{-1}\left(\beta\left(\alpha_2(\sup_i |x_i(0)|_2), k\right) + \gamma\left(\sup_i \left\|\tilde{d}_i\right\|_{\mathcal{L}_\infty}\right)\right) \tag{34}$$

$$\leq \alpha_1^{-1}\left(2\beta\left(\alpha_2(\sup_i |x_i(0)|_2), k\right)\right) + \alpha_1^{-1}\left(2\gamma\left(\sup_i \left\|\tilde{d}_i\right\|_{\mathcal{L}_\infty}\right)\right) \tag{35}$$

Let $\hat{\beta}(r,t) = \alpha_1^{-1}(2\beta(\alpha_2(r), k))$ and $\hat{\gamma} = \alpha_1^{-1}(2\gamma(r))$. We can verify that $\hat{\beta}$ and $\hat{\gamma}$ are also class-$\mathcal{KL}$ function and class-$\mathcal{K}_\infty$ function, respectively. Hence, we can then write

$$\sup_i |x_{t,i}|_2 \leq \hat{\beta}\left(\sup_i |x_i(0)|_2, k\right) + \hat{\gamma}\left(\sup_i \left\|\tilde{d}_i\right\|_{\mathcal{L}_\infty}\right) \tag{36}$$

which shows that the equilibrium point of the system is sISS.

