# OpenReview forum: "String Stable Control of Connected Vehicles via Multi-Agent Lyapunov Actor-Critic"
_ICLR.cc/2026/Conference — ICLR 2026 Conference Withdrawn Submission_

### Official Review · Reviewer_xftR · 2025-10-28

**Soundness:** 3
**Presentation:** 3
**Contribution:** 2
**Rating:** 4
**Confidence:** 4

**Summary:**

The authors propose a STRING stability criterion applicable to decision-making networks. Specifically, this criterion is a sufficient condition for scalable input-to-state stability (sISS), which provides theoretical guarantees for the STRING stability of multi-agent systems. Based on this sufficient condition, the authors further apply it to multi-agent reinforcement learning (MARL) and develop the multi-agent Lyapunov actor-critic (MALAC) algorithm. Simulation results presented by the authors demonstrate that MALAC achieves significant performance improvements compared to ACC and CACC.

**Strengths:**

1.The Theorem 1 proposed by the authors provides a sufficient condition for scalable input-to-state stability (sISS) in discrete-time multi-agent systems, thereby offering a theoretical guarantee for string stability.

2.The proposed multi-agent Lyapunov actor-critic (MALAC) algorithm is supported by certain theoretical guarantees, and simulation results further confirm its performance advantages over the adaptive cruise control (ACC) and cooperative adaptive cruise control (CACC) methods.

**Weaknesses:**

1.Theorem 1 in the manuscript, compared with Theorem 1 in Silva et al. (2025), merely appears to re-verify the proof in continuous systems within the context of discrete systems. Its contribution to the theory seems to be confined to the application level.

2.The algorithm design is somewhat simplistic. Is it overly naive to replace external disturbances with a single parameter ε? The selection of c_i,j is also rather arbitrary. In fact, different neighbors should have distinct contributions to the ego vehicle, and the application of mechanisms such as attention would appear more reasonable.

3.The baseline of the simulation experiments is too weak. The authors only compared the proposed method with early ACC/CACC approaches, which results in poor persuasiveness. It is necessary to supplement more robust baselines, including comparisons with other stability-oriented methods, to enhance the validity of the findings.

**Questions:**

Have the authors tested the proposed method in more open environments? For instance, scenarios where the platoon is not limited to a linear 1D formation but includes partial lane-changing tasks. Under the premise of such multi-leader settings, to what extent can the proposed method perform? I am particularly interested in this aspect.

---

### Official Review · Reviewer_4yzz · 2025-11-03

**Soundness:** 3
**Presentation:** 3
**Contribution:** 3
**Rating:** 4
**Confidence:** 4

**Summary:**

The paper proposes a multi-agent method that aims to mitigate and control string stability in interconnected systems. String stability, in contrast to local stability, requires robustness to disturbance on multiple parts of the network rather than only the neighbors of each node. The authors also provide a sufficient condition for string stability and propose an RL algorithm based on the conditions in the theorem to satisfy the stability. The experiments indicate very strong stability of the agents trained by the proposed method, even against multi-node disturbance.

**Strengths:**

- One of the advantages of the paper is the sound theoretical justification behind the method that directly and logically induces the RL objective.
The analytical approach, which derives stability conditions and control laws using frequency-domain analysis, Lyapunov methods, or robust control theory, is logically structured. The derivations are mathematically consistent and build clearly from assumptions to theorems.
- The generalization experiment indicates the model’s performance on unseen scenarios.

**Weaknesses:**

- The experiments are limited. Additional scenarios beyond the car platoon could be tested to evaluate the model’s performance. While the paper compares against one or two baseline controllers, it lacks comparison with recent data-driven or adaptive control frameworks (e.g., reinforcement learning-based string stabilization or model-predictive controllers). This limits the novelty claim relative to the state of the art.
- The model assumes bounded delays and symmetric bidirectional communication, which may not hold in mixed traffic or asymmetric V2V topologies.
- There is minimal exploration of parameter sensitivity, how performance changes under varying levels of delay, uncertainty, or network loss.
- Environmental and sensor uncertainties (GPS noise, actuation delay) are not integrated, which slightly narrows the scope of the work.

**Questions:**

- In the final step of devising the RL objective, the Lagrangian method transforms the constraint form into an unconstrained form. In this process, the constants on the other side of each constraint are missed (like $H_t$). How can we ensure that the optimal parameters for the unconstrained form also meet the constraints with the exact constants, because standard RL training methods don’t have such a capability?
- How can we define $c_i$ when the network is not symmetric or homogeneous? What is the relationship between $c_i$ and the structure of the network?

---

### Official Review · Reviewer_6J9h · 2025-11-03

**Soundness:** 2
**Presentation:** 2
**Contribution:** 2
**Rating:** 2
**Confidence:** 5

**Summary:**

This paper addresses the problem of maintaining string stability in networked or interconnected systems, with a motivating example in urban transportation networks (e.g., mitigating phantom traffic jams). The authors establish a sufficient condition for scalable input-to-state stability (sISS) that guarantees string stability across multi-agent systems. They derive theoretical conditions based on local Lyapunov functions that collectively ensure a global sISS property. Building on this, they introduce MALAC (Multi-Agent Lyapunov Actor-Critic), a reinforcement learning–based algorithm designed to learn control policies that preserve stability. Numerical simulations, particularly in the context of cooperative adaptive cruise control (CACC), demonstrate the method’s ability to suppress disturbance propagation and achieve stable coordination among agents.

**Strengths:**

1. The application to string stability and traffic flow control is timely and significant for both control and transportation communities.
2. The paper aims to build a theoretical grounding for the problem considered.

**Weaknesses:**

The assumptions and conditions on the nonlinear dynamics (f) are not explicitly stated.

Some of the assumptions underlying the sISS condition (e.g., on agent coupling, communication topology, or boundedness of disturbances) are not clearly stated and/or justified.

The topology is not leveraged in a meaningful way and it is unclear why the descritization of the continuous dynamics was needed?

The computational or sample efficiency aspects of MALAC are not thoroughly analyzed. It remains unclear how well the approach scales with larger networks or higher-dimensional state spaces

How the developed theory is used in the experimental setups is not clear. What is the dynamics of the examples considered in the experiment and how the sufficient conditions are used to design the controllers - these are not clearly explained.

The paper does not acknowledge a lot of works that deal with ISS (of both continuous and discrete nonlinear, interconnected systems). This is a big concern. Distributed and decentralized control of interconnected systems have been studied in control theory for several decades and it would be useful to identify the gap in the existing literature from control theory.

**Questions:**

1. What specific assumptions are made on the nonlinear dynamics $f$ and on the conditions ensuring the sISS property (e.g., coupling strength, topology, bounded disturbances)?
2. Why was discretization of the continuous dynamics necessary, and how does it influence the theoretical results or algorithm performance?
3. How is the communication topology incorporated into the analysis or controller design, and does it play a meaningful role in stability or learning?
4. Could you provide analysis or evidence on the computational and sample efficiency of MALAC, especially regarding scalability to larger networks or higher-dimensional systems?
5. How are the theoretical sufficient conditions used in the experimental setups, and what are the exact system dynamics considered in those examples?
6. How does this work relate to and differ from existing ISS-based approaches in distributed or decentralized control theory, and what specific gap does it address?

---

### Official Review · Reviewer_BHJb · 2025-11-04

**Soundness:** 2
**Presentation:** 3
**Contribution:** 2
**Rating:** 4
**Confidence:** 3

**Summary:**

The paper studies string stability in interconnected systems and proposes a sufficient condition for discrete-time scalable input-to-state stability (sISS) together with a multi-agent Lyapunov actor-critic (MALAC). The algorithm adds a Lyapunov-critic constraint that resembles the derived sISS inequality, and is evaluated on SUMO-based cooperative adaptive cruise control (CACC) scenarios with small platoons.

**Strengths:**

1. The paper is well-written and easy to follow
2. Safe multi-agent RL is an important and timely topic of research
3. The discrete-time sISS condition is easy to interpret
4. Numerical simulations show qualitative reduction in oscillations relative to ACC and CACC on small platoons.

**Weaknesses:**

That said, the paper does not meet the standards of ICLR due to issues pertaining to theory, limited evaluation (by the standards of ICLR), and incremental contribution. Furthermore, this paper might not be a good fit for ICLR and may be better considered at controls, vehicle dynamics, or transportation journals. I leave the relevance part to the area chairs but below are my technical concerns.

1. The theorem assumes inequalities on local Lyapunov functions and coupling coefficients with disturbance terms, while training uses a soft penalty with a fixed margin $\epsilon$. There is no certification that the final policy satisfies the theorem on the true closed loop with bounded disturbances. No constraint-violation statistics are reported.

2. Replacing a disturbance energy term by a constant $\epsilon$ ignores variation in disturbance magnitude. Feasibility can silently fail as conditions change, and there is no adaptive scheme.

3. Fixing $c_{i,j}=1/(|\mathcal N_i|+1)$ ensures the sum is below one but does not yield explicit slack $c>0$ or account for approximation error. No projection step enforces $\sum_j c_{i,j} \le 1 - c\Delta t$ with known slack.

4. The plots show trajectories and costs, but there is no quantitative string-stability metric such as amplification ratios, empirical $\mathcal L_2$ or $\mathcal L_\infty$ gains, frequency-response attenuation, or estimated $\beta,\gamma$ bounds.

5. The experiments are small $N$, single lane, idealized sensing and communication. No heterogeneity, sensor noise, actuation limits, V2V delays or dropouts, or mixed human drivers. No scaling to $N \ge 50$.

6. The baselines ACC and CACC are weak for comparison. Comparison with strong control baselines such as SAC or Lyapunov-constrained RL and certificate-based methods would make the paper stronger,

7. The ablation study is limited. Ablations removing the Lyapunov term, learning vs fixing $c_{i,j}$, sensitivity to $\Delta t$, entropy targets, neighbor aggregation, or penalty would tremendously improve the quality of the paper

8. The existence of suitable $V_i$ is assumed, but there is no post-training verification or fallback when the inequality is violated.

**Questions:**

I have no additional questions beyond the issues expressed in the weaknesses. A rebuttal to the weaknesses is appreciated.

---

### Note · Authors · 2025-11-21

**Comment:**

We sincerely thank the reviewers for their valuable feedback. After careful consideration, we have decided to withdraw our paper and will incorporate the comments to further improve it.

**Withdrawal Confirmation:**

I have read and agree with the venue's withdrawal policy on behalf of myself and my co-authors.